# A Study of the Relationship between Marketing and Investment in Technology Development in Transport Company

**Darius Bazaras** [1], **Margarita Išoraitė** [2] **and Kristina Vaičiūtė** [1,*]

1  Department of the Logistics and Transport Management, Vilnius Gediminas Technical University, Plytinės Str. 27, LT-10105 Vilnius, Lithuania
2  Business Management, International Business Division, Vilnius University of Applied Sciences, Didlaukio Str. 49, LT-08303 Vilnius, Lithuania
*  Correspondence: kristina.vaiciute@vilniustech.lt

**Abstract:** The article examines road transport technological development processes through the prism of marketing. Technological development alternatives of road transport companies were analysed and evaluated according to the influence of marketing on the development and competitiveness of the company's services. During the structural and technological development analysis, the marketing factors determining the components of the technological development process were determined. The analysis of scientific literature sources revealed the level of scientific problem research, which is associated with the notion that technology development is influenced by consumer demand, digital transformation, data availability, and complex processes in the logistics supply chain. When compiling the expert evaluation questionnaire, the criteria of the influence of marketing factors on the development of road transport technologies and the quality of the transport company's vehicles were singled out. The research reveals that elements of the marketing strategy of transportation companies are elastically related to personalised customer expectations and automated customer choice tracking capabilities. The results can also include quick response to troubleshooting, $CO_2$ reduction measures, ensuring process continuity, improved safety, and demand planning. The article's scientific novelty has been achieved by determining new elements of interaction between marketing and technology development in the transport company.

**Keywords:** technological development; marketing effect; transport company; digital transformation; marketing

## 1. Introduction

With the development of the country's technologies in transport, the market situation is constantly changing. In the modern world, transport companies have the opportunity to develop their activities both domestically and worldwide. Growing opportunities for transport technology, such as a range of goods or services, allow to expand the transport market and marketing and increase sales volumes and range, determining the growth of profits and setting companies' place in a particular market segment. Transport innovation exists in many developed countries as a key driver of economic growth, boosting business productivity and profitability and rapidly improving consumers' quality of life.

Keke [1] mentioned that increasing technological development brings many innovations. This situation changes communication, processes, and distribution channels between individuals and institutions. The above-mentioned technology-enriched communication systems remove time and space constraints between individuals. With extensive Internet use, social media is becoming increasingly integrated into people's lives. The spread of social networking platforms worldwide has made it possible to communicate feelings and thoughts freely. As a result, social networks have given rise to various innovations and developments in marketing science.

Lee and Trimi [2] noticed that creating a smart future requires innovative ideas to harness ubiquitous digital connectivity, smart sensors, artificial intelligence, the Internet of Things (IoT), access to all human knowledge, and entrepreneurship to take advantage of opportunities for the best quality of life. A smart future is thus one in which people are free to pursue their own well-being and to be optimistic about their future. In today's society, people are faced with many complex problems and complex solutions. According to Lee and Trimi [2], a smart future is a place where people can search for and ultimately find smart solutions to these problems in collaboration with others, with the support of governments or other entities or even on their own, using available technology and knowledge.

Cruz and Sarmento [3] noticed that new mobility models and mobility solutions are characterized by greater flexibility in exploiting the advantages of the "sharing concept" and at the same time provide solutions that emit less greenhouse gas (GHG). According to the authors, these dynamics and the evolving environment present several new challenges at different levels, promoting the development of mobility as a service (MaaS). Wangai, Rohacs, and Boros [4] mentioned that future technologies will improve efficiency and safety and reduce the environmental impact of future transport systems. The study of these technological effects will develop general methods at the system level and specific methods and subsystems at the vehicle level.

Taking into account the above, the purpose of this paper is the study of the relationship between marketing and investment in technology development in transport. The peculiarity of this research is that the processes taking place in the transport company are individually designed and evaluated in the market. Hence, a large part of the investment needs to be allocated to the development of the technology of the transport company to ensure participation in the transport market, and this is greatly influenced by marketing. The novelty of the research is that in the activities of the transport company, there is an intensive assessment of the rhythm of service provision and looking at the possibilities of technology development through a marketing approach and using a multi-criteria method, which allows to scientifically justify the need to use innovative technologies and make effective marketing management decisions.

Further material is divided into several parts. Thus, Section 2 presents a scientific literature review about technological development in the transport company and its marketing impact on technological development. Section 3 presents the research methodology and methods. Section 4 presents empirical data analysis. Section 5 presents a discussion of the results presented by this article's authors. Section 6 presents conclusions.

## 2. Literature Review

### 2.1. Technological Development in the Transport Company

Technological development is a crucial area in the freight transport process. The technology could generally be described as a real-time freight process where disruptions are promptly identified and rectified before the customer notices. Intelligent technologies are associated with the speed of product delivery and the continuity of the freight process. The technology concept has long been mentioned in foreign countries. Intelligent technologies (IT) are difficult to define clearly, but they are understood as the application of information and communication technologies in the transport field.

The Policy Department for Structural and Cohesion Policies Directorate-General for Internal Policies [5] noticed that various new technologies (such as smartphones, sensors, block circuits, and artificial intelligence) drive innovation in smart mobility. Rising pressure to achieve public goals in the transport sector (e.g., reducing carbon dioxide, improving road safety, congestion reduction) will be another factor in developing smart mobility. However, there are still many challenges to deploying smart mobility applications; in this way, it increases the benefits for Europe and, at the same time, reduces any negative effects. The disadvantages of harmonising national legislation and the lack of social opposition are just two examples of the problems that may interfere with their large-scale deployment. Therefore, on the other hand, it can be assumed that the attractiveness of new technologies

available to users simultaneously encourages the development and wider application of these technologies in practice. The aspect of wider applicability is one of the driving forces behind developing new technologies.

Kolasińska-Morawska, Sułkowski, and Morawski [6] mentioned that the level of competitiveness of modern economies depends on the implementation and dissemination of innovations based on new technologies. Artificial intelligence, Internet of Things (IoT), hyper connectivity, cloud computing applications and services, big data analytics (BDA), big data as a service (BDaaS), automation, and robotics are just a few of the technologies that the authors considered worth exploring in more detail transport. All the new information and communication technologies mentioned here—big data, cloud data arrays, and real-time factor actualisation—are associated with the very important concept of mobility, which, by its very nature, is inseparable from the activities of transport and logistics companies. Digital transformation is also a new opportunity and challenge for businesses.

Vaičiūtė and Bureika [7] investigated technological development (experimental, development or pilot, construction, and technological works)—acquisition, combination, and formation and application of scientific, technological, business, and other knowledge and skills for the implementation plans, layout (technological) of new or improved products, processes, or services to create schemes or models.

Klein, Gutowska, and Gutowski [8] noticed that although for some transport and logistics spaces, such as warehouse management, it has been said that "The future has already come", as solutions that seem completely independent and have become very efficient as a standard, there were sectors severely affected by the shortage innovation and digitisation. One of them was the supply chain, which lacks complete control of the flow of goods due to insufficient and incompatible digitisation between partners and has resulted in high-cost generation and cannot achieve many benefits, such as greater customer experience, faster product delivery, opportunity to respond quickly to emerging threats, building loyalty relationships with the customer, and so on. According to the authors, innovation is necessary to respond effectively to the economic devastation caused by the global pandemic. In the T&L sector, not only aspects related to new technologies but also business management rules and organisational, product, and process innovations need to be transformed to keep pace with technological innovation. The problem is multidimensional, and the complexity of the T&L problem is because different elements of it are further developed with different priorities, flexibility, and ease of change. The most important is sustainability policy, i.e., ensuring sustainable and green growth based on innovation. In any case, we are dealing with very complex processes related to logistics supply chain management issues, which constantly require updating, innovation, and optimisation. In the logistics supply chain, there are many technological processes, the success of which directly affects the success of the final result—the satisfaction of the user's needs.

Kostrzewski, Filina-Dawidowicz, and Walusiak [9] mentioned that logistics centres face various challenges in developing technical and technological facilities. On the one hand, new solutions have several advantages in implementation. On the other hand, logistics centres face threats that can disrupt modern deployment technologies. In addition, the technological development of these centres and the areas of new solutions may differ. "Implementation may vary depending on the geographical region and economic environment. Therefore, the current operating condition and needs of such logistics facilities and viable solutions should be identified and should be analysed in detail. Before performing any analysis in companies, review the literature on the technologies currently used in logistics facilities have been made." Here, the authors touch on a very relevant problem—further technological development in different regions, companies, and economic environments. The different levels of action and the different complexity of the technologies used are a significant problem in achieving the synchronisation of activities and solving the problem of efficiency.

Babaleye and Greblikaite [10] noticed that smart technologies can be deployed even in not-digitised cities; however, some enabling measures need to be implemented. First,

technology is needed to enable the use of ICT operation so that users can access it. As the smartphone evolved, the primary access mode of ICT was firmly entrenched. Second, there must be a certain level of consumer acceptance. Although the first condition can now be easily fulfilled by using smartphones and internet connection, which is a daily feature of city life, the second is less obvious, especially for transport consumers. Research shows that there seems to be a surprisingly large group of consumers who are not interested in any form of smart technology. However, analysing the authors' insights, one can notice an interesting potential dependency—possibly, technological development is possible and encouraged through mobile phones, tablets, or other personal devices. Although the user denies their interest in smart technologies, the mobile devices they use, and especially the software they contain, provide the conditions for engagement and development.

Liu, Wu, and Chu [11] stated that to measure technological change and environmental performance accurately and to improve further the technology and environmental performance of the road transport industry in the context of "big data", they proposed a Hicks–Moorsteen index model based on DEA and then implemented it to assess the performance of the road transport industry.

Harris, Wang Y., and Wang H. [12] mentioned that the role of information and communication technologies (ICT) in freight transport is well-known as a key tool. However, recent advances in ICT in multimodal freight transport in the United Kingdom and Europe have been slow. Harris, Wang Y., and Wang H. [12] investigated the possible reasons for such slow adoption and assessed how the latest technological advances, such as cloud computing and the Internet of Things, may have changed the landscape and thus helped to overcome these barriers.

Lind and Melander [13] mentioned that technological developments rapidly affect the road freight transport system. In parallel, new business models were proposed that bring them closer to the network and complex business features. Lind and Melander [10] raised the question of what this means for the road haulage business from the point of view of the truck manufacturer. The aim was to analyse the content and development of Swedish road freight transport network business models to contribute to sustainable transport solutions.

Maheshwari and Axhausen [14] examined whether a technological change in transport will lead cities to the path of sustainability in five aspects—traffic flow, use of space, energy consumption, transit and active mobility and affordability—in the review of 34 quantitative studies. The authors found that these studies can provide more accurate answers based on analytical and simulation models. Their results contradict each other depending on the starting conditions, simulation methods, and other driving factors. These drivers fall into four categories: technological integration, policy, operations, and urban planning.

Summing up the analysis of the scientific works reviewed in this section, it can be said that we are dealing with a sufficiently complex problem field, which includes the importance of technology development in transport companies and their impact on the areas of activity. It is noticeable that the practical application of new technologies encourages the development of new technologies. In this context, it has also been noticed that new technologies are increasingly focused on mobility, in a broad sense, which is especially relevant for transport companies. The complexity of transport and logistics processes, the relevance of logistics supply chain management, and the impact on the environment are also noticeable in the analysed literature sources. It can be assumed that technological development would be affected by rapid troubleshooting; adaptation to the speed of delivery of goods; ensuring the continuity of the cargo transportation process; the reduction of carbon dioxide, improvement of road safety, and reduction of traffic jams; the implementation of digital transformation; and the use of big data for statistical analysis.

## 2.2. The Relationship between Marketing and Technological Development

Schmidt [15] mentioned that the future of technology promises to replace traditional marketing functions. Several of these technologies and their impact on the marketing function are reviewed and disaggregated to understand the benefits they can provide; the

most important are decision-support technologies that lag far behind demand planning and sales team management tools. Specifically, to what extent do technologies crowd out the "set of disciplines and responsibilities" embedded in classic marketing functions? How is the function being replaced by the highest-quality, demand-driven processes based on available technologies that allow individuals to achieve what the whole "set of institutions" has tried to accomplish? These technologies are available today, and companies that adopt these technologies will have a clear competitive advantage, ensuring that customer expectations are met in a more personalised, customised process more efficiently and to the benefit of both investors and investors.

Gupta, Natarajan, and Raghuram [16] discussed the impact of technology on the marketing techniques used today. One way would be to review online marketing using big data research techniques with data mining that changed the marketing world and gave market researchers an extensive data set for accurate analysis and formulation of strategies. In addition, social media has replaced market research in many ways and created an environment that gives unfiltered feedback. Today's technology has changed the scenario of a customer relationship strategy. These technologies include improved search services, increased computing speed, and e-commerce.

Cham et al. [17] mentioned that in all aspects of business, the emergence of new technologies such as artificial intelligence, big data, blockchain, virtual reality, and robots has created a new paradigm shift and led to innovation in marketing research and practice. Such a transformation has become a marketing catalyst, perpetuating new marketing trends and archetypes in digital marketing and marketing analytics. In recent years, the advancement of digital marketing through social media has grown beyond the original goal of a social networking platform. Instead, it has become a platform that allows companies to interact with customers almost instantly and participate directly in developing marketing strategies.

Peyravi, Nekrošienė, and Lobanova [18] noticed that artificial intelligence is an increasingly popular term. This is particularly appealing to many researchers in the field of marketing; however, it lacks a specific definition. Technically, artificial intelligence can be defined as an interactive process between robots, computers, cloud computing, networking, and digital content production in a variety of day-to-day operations, especially in businesses.

Hoffman et al. [19] mentioned that new technologies have often been effectively implemented to improve business-to-consumer interaction through new marketing tools. For example, artificial intelligence is a powerful engine that replaces human company representatives with machine agents, facilitating company-user interaction through the "machine world". Anthropomorphised chat robots can influence user response to user-initiated service interactions. In addition, pseudo-portraits are increasingly used in business-to-consumer interactions, where the scale of the pseudo-portrait form and the reality of its behaviour are key factors in its effectiveness.

Jain and Yadav [20] noticed that marketing combines art and applied science and uses information technology. Marketing is applied in companies and organisations through marketing management. Best-input result technology was an innovation. Today's rapid innovation is the key to success. A storm hit products on the market and the audience. Many products are on the market to meet the customers' needs and market desire. Needs, whether basic, hidden, or dreamed, keep marketers on their feet. "Technology for marketing" is a myth; although technology offers many advantages, it has many negative points. The benefits of technology attract new customers, help automate tracking, help participation in online decision making, etc. On the other hand, there are difficulties dealing with the shortness of time.

Berdiyorov [21] mentioned that the country had made great strides in developing marketing innovations in the past. In the Republic of Uzbekistan, much attention is paid to implementing innovations in transport. In particular, in 2017, 2046 innovations were introduced in all sectors of the economy, resulting in more than 15 jobs and trillions of

amounts of innovative products. Twenty-three innovations were introduced in transport and storage in 2017, and their share in total innovation amounted to 3.3 per cent. It is noteworthy that in 2015–2017, only three marketers' innovations in transportation and storage were introduced. While the introduction of marketing innovation in many industries grew rapidly during the period under review, a trend behind innovation in the transport sector is declining.

Leonow et al. [22] stated that in today's world, information technology is used for both market research and advertising of goods and services (in the broadest sense, also for image purposes). Using market research is key information marketing technologies in the current stage of development, which include online surveys, online advertising, a set of geographic surveys, and statistical analysis of big data. In the use of big data and the technologies already mentioned, viral advertising uses videos that combine humour, fun, and teaching. An action analysis is also performed by site users who help create an individual offer for each customer. It also increases advertising effectiveness for its personalisation, such as displaying different types of ads depending on the interests of the customers. The company can more easily manage its development, defining the demand for a good.

Hosseini, Mohammadi, and Safari [23] mentioned that the development of communication technologies for satellites, computers, cable TV, video conferencing, and computer networks allowed people to obtain information faster. Therefore, it is important to study the impact of technology on different market shares and marketing. Current study findings show that the new technology and its entry into the marketing field affected various aspects of the marketing mix. According to marketing experts, the combination of advertising is the first part that was affected by using new technologies. The major role of advertising is to present a wide range of goods and thus strengthen the market economy. Accordingly, the impact of this part of the technology as a result and relevance is influenced by various aspects of the marketing complex, such as product, price, and distribution.

Zhu [24] mentioned that businesses need to adapt to the new business environment with digital technology in the digital age. Both digital platforms and big data can help companies make decisions when deciding on their marketing strategy. However, before digital technologies were more widely used than digital platforms, big data could contribute to a business's marketing strategy decisions. The event and use of digital platforms and big data changed the way the company markets. With such an impact, it is important to understand how digital platforms and big data can affect businesses' marketing strategies.

Shugan [25] noticed that there are marketing issues related to emerging marketing technologies. With the advent of new and different technologies, the potential benefits are being reaped as the basis of any marketing campaign—they could not occur immediately and eventually develop in fundamentally different directions. Most failed technologies do this because they do not add value benefits that arise. Thus, the old, boring marketing concept based on consumer benefits continues to be stolen by the charm of many gee-whiz technologies. Maybe using tracking tools to discover new benefits can defeat traditional marketing research tools for company research and development.

Thus, technological developments affect all elements of the marketing mix. However, it is most noticeable in the areas of product development, improvement, and production. Each major scientific achievement can become the reason for new goods' appearance, creating new needs. Modern information technologies have revolutionised marketing processes, creating new forms in the labour market and fundamentally changing the ways of communicating with market participants. The needs of marketing determine the improvement of technological processes. It can be said that the interaction between the marketing mix and technological change is mutual. Marketing activities stimulate the customers' and users' needs for new technologies by creating or—what is much more critical—inspiring user expectations for new product quality and functionality. A comprehensive information field creates favourable circumstances for users to quickly receive information about changes in the market, alternative products, and newly created added value of the product. The existing technological breakthrough, innovation, or new technology used affects the mar-

keting complex, enriching it with new concepts, attitudes, values, and forms of expression focused on emphasising innovation.

After analysing the sources of the scientific literature presented in this section, it can be said that a "vicious circle" may exist—marketing promotes technologies and their wider use, and the development of technologies changes marketing and shapes it in a new way, which helps to develop new technologies again. Innovations in marketing appear in the developing use of big data and cloud data arrays, which allows market research in a completely different way to determine the mechanisms of impact on the user and the means of market formation. There is a noticeable possible trend that the marketing will not be so directly tied to a person or a classic survey but will be connected with a great deal of information from big data or cloud data arrays, which will form from consumer devices, vehicle monitoring or tracking devices, mass consumption, or behavioural analysis tools. As it is worth assuming, artificial intelligence tools will be used more and more in marketing, both focused on the user and optimising or managing internal marketing processes. Based on the opinion of the scientists mentioned above, the impact of marketing on technological development can be manifested through the statistical analysis of big data, the possibilities of applying big data research methods, and established loyalty relationships with the customer; feedback; demand planning, a more personalised process that meets customer expectations; and sales team management tools to automate the tracking of customer choices.

## 3. Methodology and Methods

A multi-criteria approach was chosen to assess the impact of marketing on the technological development of the transport company and the intensity of the technological development and the quality assurance of the transport company's services. Based on the multi-criteria method, the expert group $n$ quantifies $m$ objects. Estimates form a matrix of $n$ rows and m columns in Table 1 [26].

**Table 1.** Matrix of evaluation indicators (compiled by the authors).

| Expert Code | | Factor Encryption Symbol, $j = 1, 2, \ldots, m$ | | | | |
| --- | --- | --- | --- | --- | --- | --- |
| | | $X_1$ | $X_2$ | $X_3$ | $\ldots$ | $X_m$ |
| $i = 1, 2, \ldots, n$ | $E_1$ | $R_{11}$ | $R_{12}$ | $R_{13}$ | $\ldots$ | $R_{1m}$ |
| | $E_2$ | $R_{21}$ | $R_{22}$ | $R_{23}$ | $\ldots$ | $R_{2m}$ |
| | $E_3$ | $R_{31}$ | $R_{32}$ | $R_{33}$ | $\ldots$ | $R_{3m}$ |
| | $\ldots$ | $\ldots$ | $\ldots$ | $\ldots$ | $\ldots$ | $\ldots$ |
| | $E_n$ | $R_{n1}$ | $R_{n2}$ | $R_{n3}$ | $\ldots$ | $R_{nm}$ |
| $\sum_{i=1}^{n} R_{ij}$ | | $\ldots$ | $\ldots$ | $\ldots$ | $\ldots$ | $\ldots$ | $\ldots$ |
| $\overline{R}_j = \frac{\sum_{i=1}^{n} R_{ij}}{n}$ | | $\ldots$ | $\ldots$ | $\ldots$ | $\ldots$ | $\ldots$ | $\ldots$ |
| $\sum_{i=1}^{n} R_{ij} - \frac{1}{2}n(m+1)$ | | $\ldots$ | $\ldots$ | $\ldots$ | $\ldots$ | $\ldots$ | $\ldots$ |
| $\left[ \sum_{i=1}^{n} R_{ij} - \frac{1}{2}n(m+1) \right]^2$ | | $\ldots$ | $\ldots$ | $\ldots$ | $\ldots$ | $\ldots$ | $\ldots$ |

Indicator units, unit parts, and percentages can be evaluated in a ten-point system. The ranking of expert indicators is suitable for calculating the concordance coefficient. The ranking is a procedure in which the most critical indicator is given a rank of $R$, equal to one, the second indicator is given a second rank, and the last indicator is given a rank of $m$ ($m$ is the number of benchmarks). The average of the sum of ranks is calculated [27]:

$$\sum_{i=1}^{n} R_{ij} = \frac{1}{2}n(m+1) \tag{1}$$

where $R_{ij}$—a rank of $R$; $m$—the number of benchmarks; $n$—the number of experts.

$$W = \frac{12S}{n^2 m(m^2 - 1)} = \frac{12S}{n^2(m^3 - m)} \tag{2}$$

where $W$—the concordance coefficient; $S$—the sum of the squares of the deviation from the arithmetic mean.

$$\chi^2 = n(m - 1)W = \frac{12S}{nm(m + 1)} \tag{3}$$

where Pearson criterion—$\chi^2$.

$$W_{\min} = \frac{\chi^2_{v,\alpha}}{n(m - 1)} \tag{4}$$

The expert evaluation indicators' opinions' consistency is determined by calculating the concordance coefficient of the Kendall ranks. Suppose $S$ (variance) is the real sum of the squares calculated according to formula (1). In that case, the concordance coefficient (2), in the absence of related ranks, is defined by the ratio of the resulting $S$ to the corresponding maximum $S_{\max}$. The cut-off value of the concordance coefficient ($W$) is determined when the expert assessments can be considered harmonised, and the significance of the concordance coefficient can be determined using (3) the Pearson criterion $\chi^2$. The minimum value of the concordance coefficient $W_{\min}$ is calculated (4).

*The research process is based on the statements (hypotheses) (a) the technological development strategy of the transport company depends on the market trend; (b) the marketing strategy influences the technological development of Lithuanian transport companies and their intensity of investment.* Given that the minimum recommended number of experts for forming the expert group is 3, and the optimal size of the expert group is from 8 to 10 experts, 8 experts were included [28,29].

The competency coefficient of each expert is calculated using the method of calculating the expert competence coefficient [30]. In formula (5), all experts are given the same competence coefficient [31]. Giving the same weight to all experts shows whether the views of the experts are unanimous and competent. For this purpose, the competence factor of experts is calculated:

$$K_j^0 = \frac{1}{n}, j = 1, \ldots, n, \tag{5}$$

here, $n$—number of experts; $K_j{}^0$—expert competence coefficient; $j$—coefficient = 1.

The sums of the initial rank values in the columns are then multiplied by the initial competency coefficient. Group estimates of alternatives (6) and a new matrix for calculating the competence factor were obtained. The competence coefficient [31] is calculated according to Formulas (5)–(10):

$$X_j^t = \sum_{i=1}^{m} K_i^{t-1} \cdot x_{ij}, \ j = 1, \ldots, n, \tag{6}$$

$X_j^t$—new matrix values; $\sum_{i=1}^{m} K_i^{t-1}$—group assessments; $x_{ij} = i$—experts; $j$—the rank of the alternative.

$$\lambda^t = \sum_{j=1}^{n} \sum_{i=1}^{m} x_j^t \cdot x_{ij}, \tag{7}$$

$\lambda^t$ lambda—all matrices; $x_j^t$—the sum of the values; $n$—number of experts; $m$—number of alternatives.

$$K_i^t = \frac{1}{\lambda^t} \cdot \sum_{j=1}^{n} x_j^t \cdot x_{ij}, \sum_{i=1}^{m} K_i^t = 1. \tag{8}$$

In the direct method weighting of criteria, $c_{ik}$, the sum of the weights of all the evaluations of each expert must be equal to one (or 100%). The method used to determine the criteria weights indirectly uses the chosen scoring system (5, 10, 20, etc.). Evaluations

may be repeated. The direct and indirect estimation methods calculate the weights $w$ of the criteria according to the formula [27]:

$$w = \frac{\sum_{k=1}^{r} c_{ik}}{\sum_{i=1}^{m} \sum_{k=1}^{r} c_{ik}}. \tag{9}$$

Expert evaluations are noted as $c_{ik}$ ($i = 1, \dots, m; k = 1, \dots, r$), where $m$—the number of applied criteria, and $r$—the number of experts.

Several logistics specialists from Lithuanian logistics and transport companies participated in the study. One of the essential characteristics of experts is competence, so experts were required to have competence and experience in the field under investigation. Everyone must have a university degree and at least five years of managerial work experience in developing a transportation service. All experts have 5–8 years of management experience in Lithuanian logistics and transport companies and higher education.

The experts were asked several questions, based on which it was necessary to evaluate the importance of the transport company's marketing criteria for the implementation of technological development and to evaluate the factors influencing the transport company's investments in technological development. Experts were asked to confirm or deny the statement that the technological development strategy of a transport company depends on market trends.

## 4. Empirical Analysis

The study involved eight experts from different transport enterprises. The experts had 5 to 8 years of experience in managing logistics and transport enterprises.

Experts had to use a multi-criteria ranking method to assess the importance of the criteria (in order of importance: 1—most important, 8—least important) for the transport company's intensity of investment in the creation and development of technological infrastructure, which is presented in Figure 1:

(a)  For quick troubleshooting;
(b)  Adjust for speed of product delivery;
(c)  To ensure the continuity of the cargo transportation process;
(d)  To reduce carbon dioxide;
(e)  To improve road safety;
(f)  Reduce congestion;
(g)  For the introduction of digital transformation;
(h)  For the performance of statistical analysis of big data.

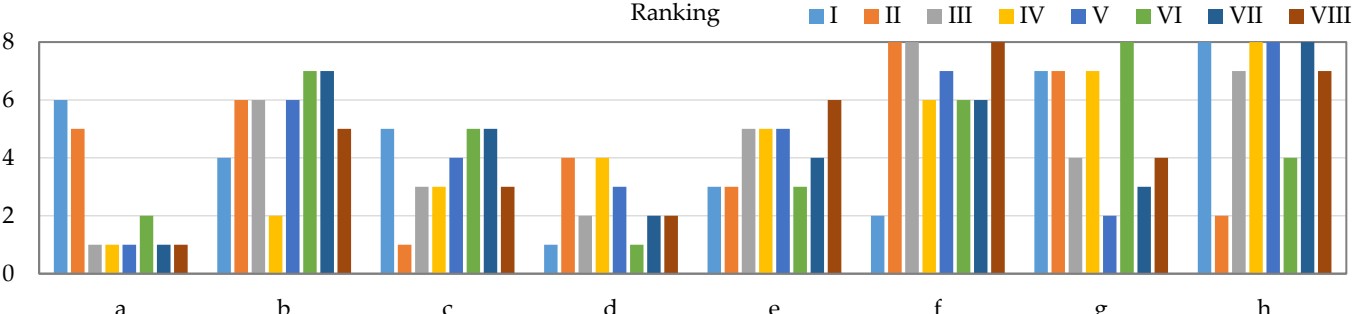

**Figure 1.** Distribution of expert (I–VIII) ranks of the criteria for the intensity of investments of the transport company in the creation and development of technological infrastructure (compiled by the authors).

The analysis data and the calculation of the distribution of the rankings from the questionnaires by eight experts are listed in Table 2.

**Table 2.** Companies ranking the importance of the criteria for the intensity of investments of the transport company in the creation and development of technological infrastructure (compiled by the authors).

| Respondent's Queue. No. | Factor Encryption Symbol ($m = 8$) | | | | | | | |
|---|---|---|---|---|---|---|---|---|
| | **a** | **b** | **c** | **d** | **e** | **f** | **g** | **h** |
| $\sum_{i=1}^{n} R_{ij}$ | 18 | 43 | 29 | 19 | 34 | 51 | 42 | 52 |
| $\overline{R}_j = \frac{\sum_{i=1}^{n} R_{ij}}{n}$ | 2.25 | 5.375 | 3.625 | 2.375 | 4.25 | 6.375 | 5.25 | 6.5 |
| $\sum_{i=1}^{n} R_{ij} - \frac{1}{2}n(m+1)$ | −18 | 7 | −7 | −17 | −2 | 15 | 6 | 16 |
| $\left[\sum_{i=1}^{n} R_{ij} - \frac{1}{2}n(m+1)\right]^2$ | 324 | 49 | 49 | 289 | 4 | 225 | 36 | 256 |

The concordance coefficient $W$ is calculated according to formula (2) when there are no associated ranks.

$$W = \frac{12S}{n^2(m^3 - m)} = \frac{12 \times 1232}{8^2(8^3 - 8)} = 0.4583.$$

The number of important criteria ($m$) for the influence of the technological development of a transport company on the implementation of the intensity of investments is 8, i.e., $m > 7$. Then, the weight of the concordance coefficient $\chi^2$ is calculated according to formula (3), and a random quantity is obtained.

$$\chi^2 = n(m-1)W = \frac{12S}{nm(m+1)} = \frac{12 \times 1232}{8 \times 8(8+1)} = 25.667.$$

$\chi^2$, the estimated value of 25.667, was higher than the critical value (equal to 14.0671), which is why the respondents' opinion is considered consistent, and the average ranks show the general opinion of the experts.

$$W_{\min} = \frac{\chi^2_{v,\alpha}}{n(m-1)} = \frac{14.0671}{8(8-1)} = 0.2511 < 0.4583.$$

The lowest value of the concordance $W_{\min}$ coefficient, calculated according to formula (4), where $W_{\min} = 0.2511 < 0.4583$, states that the opinions of all eight respondents on the eight main criteria of technological development of a transport company, which are important for the intensity of investments of the transport companies, are still considered harmonised.

The leading indicators of the importance of the technological development of the transport company, which are important for the implementation of the intensity of investments of the transport company in the creation and development of technological infrastructure, were calculated—$Q_j$. The obtained data are presented in Table 3.

**Table 3.** The main important indicators of the importance of technological development of the transport companies' intensity of investments in the creation and development of technological infrastructure—$Q_j$ (compiled by the authors).

| Indicator Marker | Factor Encryption Symbol ($m = 8$) | | | | | | | | Sum |
|---|---|---|---|---|---|---|---|---|---|
| | **a** | **b** | **c** | **d** | **e** | **f** | **g** | **h** | |
| $q_j$ | 0.0625 | 0.149306 | 0.100694 | 0.065972 | 0.118056 | 0.177083 | 0.145833 | 0.180556 | 1 |
| $d_j$ | 0.9375 | 0.850694 | 0.899306 | 0.934028 | 0.881944 | 0.822917 | 0.854167 | 0.819444 | 7 |
| $Q_j$ | 0.133929 | 0.121528 | 0.128472 | 0.133433 | 0.125992 | 0.11756 | 0.122024 | 0.117063 | 1 |
| $Q_j{}'$ | 0.1875 | 0.100694 | 0.149306 | 0.184028 | 0.131944 | 0.072917 | 0.104167 | 0.069444 | 1 |
| Factor layout | 1 | 6 | 3 | 2 | 4 | 7 | 5 | 8 | |

Based on expert assessments and calculations, the list of importance of the main criteria for the impact of the technological development of the transport companies' intensity of investments in the creation and development of technological infrastructure should be arranged in the following order, and the five main ones are presented:

For quick troubleshooting;

To reduce carbon dioxide;

To ensure the continuity of the cargo transportation process;

To improve road safety;

For the introduction of digital transformation.

All experts (100%) confirmed the statement that the technological development strategy of the transport company depends on the existing market trends.

Experts used the multi-criteria ranking method to assess the importance of the criteria (in order of importance: 1—most important, 8—least important) of the marketing of the transport company for the implementation of the technological development, which are presented in Figure 2:

(a)  For statistical analysis of big data;

(b)  The possibility of applying big data research methods;

(c)  Establish a loyalty relationship with the customer;

(d)  An environment was created to provide unfiltered feedback;

(e)  For demand planning;

(f)  A more personalised customised process to meet customer expectations;

(g)  For sales team management tool;

(h)  To help automate the tracking of customer choices.

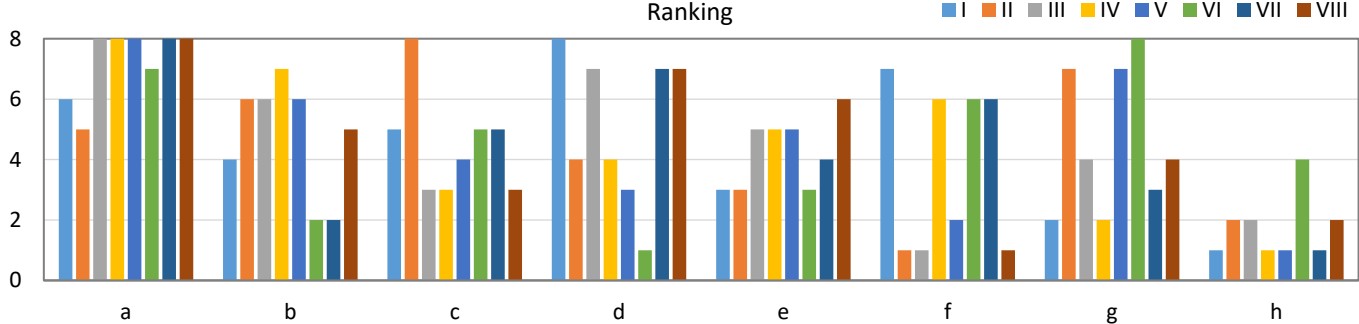

**Figure 2.** Distribution of expert (I–VIII) rankings of the influence of the marketing of the transport company for the implementation of the technological development (compiled by the authors).

The data of the analysis and calculation of the distribution of the rankings of the questionnaires of the eight experts are listed in Table 4.

**Table 4.** Ranking table of the main importance criteria of the influence of the marketing of the transport companies for the implementation of the technological development (compiled by the authors).

| Respondent's Queue. No. | Factor Encryption Symbol ($m = 8$) | | | | | | | |
|---|---|---|---|---|---|---|---|---|
| | **a** | **b** | **c** | **d** | **e** | **f** | **g** | **h** |
| $\sum_{i=1}^{n} R_{ij}$ | 58 | 38 | 36 | 41 | 34 | 30 | 37 | 14 |
| $\overline{R}_j = \frac{\sum_{i=1}^{n} R_{ij}}{n}$ | 7.25 | 4.75 | 4.5 | 5.125 | 4.25 | 3.75 | 4.625 | 1.75 |
| $\sum_{i=1}^{n} R_{ij} - \frac{1}{2}n(m+1)$ | 22 | 2 | 0 | 5 | −2 | −6 | 1 | −22 |
| $\left[\sum_{i=1}^{n} R_{ij} - \frac{1}{2}n(m+1)\right]^2$ | 484 | 4 | 0 | 25 | 4 | 36 | 1 | 484 |

The concordance coefficient $W$ is calculated according to Formula (2) when there are no associated ranks.

$$W = \frac{12S}{n^2(m^3 - m)} = \frac{12 \times 1038}{8^2(8^3 - 8)} = 0.3861.$$

The number of important criteria ($m$) for the influence of a transport company's technological development on the marketing strategy's implementation is 8, i.e., $m > 7$. Then, the weight of the concordance coefficient $\chi^2$ is calculated according to Formula (3), and a random quantity is obtained.

$$\chi^2 = n(m-1)W = \frac{12S}{nm(m+1)} = \frac{12 \times 1038}{8 \times 8(8+1)} = 21.625.$$

$\chi^2$, the estimated value of 21.625, was higher than the critical value (equal to 14.0671), which is why the respondents' opinion is considered consistent, and the average ranks show the general opinion of the experts.

$$W_{\min} = \frac{\chi^2_{v,\alpha}}{n(m-1)} = \frac{14.0671}{8(8-1)} = 0.2511 < 0.3861.$$

The lowest value of the concordance $W_{\min}$ coefficient, calculated according to Formula (4), where $W_{\min} = 0.2511 < 0.3861$, states that the opinions of all eight respondents on the eight main criteria of marketing, which are important for the implementation of technological development of a transport company, are still considered harmonised.

The main indicators of the importance of the marketing of the transport company, which are important for the implementation of the technological development, are calculated—$Q_j$. The obtained data are presented in Table 5.

**Table 5.** Significance indicators $Q_j$ of the main technological development of the transport company, which are important for the implementation of the marketing strategy (compiled by the authors).

| Indicator Marker | Factor Encryption Symbol ($m = 8$) | | | | | | | | Sum |
| --- | --- | --- | --- | --- | --- | --- | --- | --- | --- |
| | **a** | **b** | **c** | **d** | **e** | **f** | **g** | **h** | |
| $q_j$ | 0.201389 | 0.131944 | 0.125 | 0.142361 | 0.118056 | 0.104167 | 0.128472 | 0.048611 | 1 |
| $d_j$ | 0.798611 | 0.868056 | 0.875 | 0.857639 | 0.881944 | 0.895833 | 0.871528 | 0.951389 | 7 |
| $Q_j$ | 0.114087 | 0.124008 | 0.125 | 0.12252 | 0.125992 | 0.127976 | 0.124504 | 0.135913 | 1 |
| $Q_j{}'$ | 0.048611 | 0.118056 | 0.125 | 0.107639 | 0.131944 | 0.145833 | 0.121528 | 0.201389 | 1 |
| Factor layout | 8 | 6 | 4 | 7 | 3 | 2 | 5 | 1 | |

Based on expert assessments and calculations, the list of importance of the main criteria for the impact of the marketing strategy of a transport company on the implementation of technological development should be arranged in the following order, and the five main ones are presented:

- Helps automate the tracking of customer choices;
- A more personalised customised process to meet customer expectations;
- For demand planning;
- Establish a loyalty relationship with the customer;
- For sales team management tools.

The calculated Kendall concordance coefficient does not identify those experts whose evaluations may differ from others. The competence coefficient is calculated using Formulas (5)–(9).

In this respect: $K_j^0 = \frac{1}{8} = 0.125$. The sums of the initial values in the columns of Table 5 are then multiplied by the initial competency coefficient. Group estimates of alternatives (6) and a new matrix for calculating the competence factor were obtained. To calculate the final

Kendall expert competence coefficients, the sum of each row of the matrix is divided by lambda (7), the size of which is 2089. It is important to note that the sum of the competence estimates thus calculated must be equal to one. According to the analysis and the obtained results in Table 6, it can be stated that the 8th and 7th experts have the highest (equal) competence in comparison with all the experts who participated in the survey.

**Table 6.** Competence coefficients of experts (compiled by the authors).

| Expert Competence Coefficients | | | | | | | |
|---|---|---|---|---|---|---|---|
| E1 | E2 | E3 | E4 | E5 | E6 | E7 | E8 |
| 0.1376 | 0.1077 | 0.1258 | 0.1321 | 0.1162 | 0.0962 | 0.1415 | 0.1426 |

To check that all experts are competent, we calculated according to the formula $\overline{K}_i^t - 1.96s \leq K_i^t \leq \overline{K}_i^t + 1.96s$ $\overline{K}_i^t$ the average of the competence coefficients; $s$ is the standard deviation, and we obtained intervals [0.027; 0.223]. The competence of the 1st expert in this group of experts is the lowest (0.0962) but not so low that the expert assessment should be eliminated during the research. In summary, it can be stated that the experts with the highest length of service in managerial positions for more than five years and the same coefficients of competence were 0.1426. Other experts had a sufficient level of competence for their assessments to be taken into account.

Taking into account the above, it is possible to propose a model of marketing impact on the technological development of transport companies. This model is shown in Figure 3.

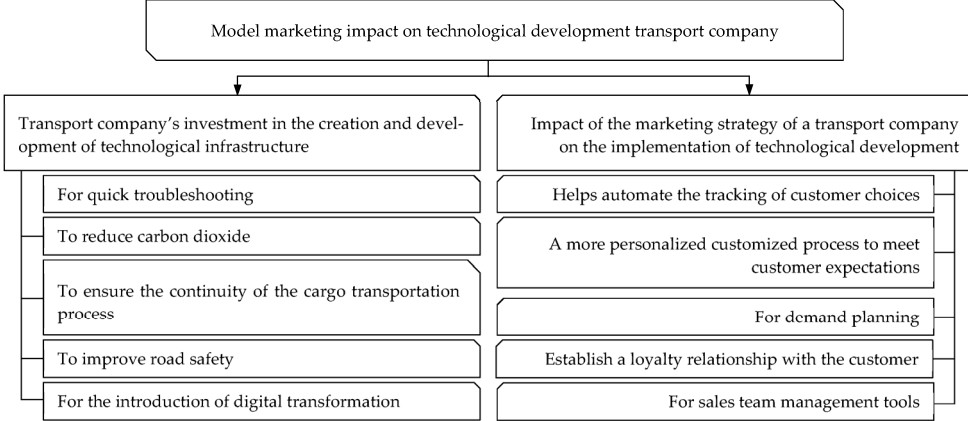

**Figure 3.** Model of marketing impacts the technological development of transport companies.

As can be seen from Figure 3, the model of marketing impact on the technological development of transport company consists of a transport company's investment and impact of the marketing strategy of a transport company on the implementation of technological development.

## 5. Discussion

Recently, there has been increasing interest in the impact of marketing on technological development in transport companies. In order to determine the importance of the criteria for the transport company's investment in the creation and development of technological infrastructure and the importance of the criteria of the marketing of the transport company for the implementation of the technological development, a study was conducted. Several authors researched the effects of marketing on technological development in transport companies, such as [4,7,14,21,32,33].

This article's authors' hypothesis results are based on an assessment of the importance of the criteria for the transport company's investment in the creation and development of technological infrastructure through quick troubleshooting; adjusting for speed of product delivery; ensuring the continuity of the cargo transportation process; reduction of carbon

dioxide; improvement road safety; reduction of congestion; for the introduction of digital transformation; and for the performance of statistical analysis of big data.

In this article, the authors for hypotheses confirmed assessed the importance of the criteria of the marketing of the transport company for the implementation of technological development: for statistical analysis of big data; the possibility of applying big data research methods; establishing a loyal relationship with the customer; creating an environment to provide unfiltered feedback; for demand planning; creating a more personalised customised process to meet customer expectations; for a sales team management tool; and helping automate the tracking of customer choices.

Both hypotheses were proven to be correct and were confirmed.

The advantage of the performed research is that the authors of this article found that elements of marketing and technological infrastructure must be oriented towards a more personalised, customised process that meets customer expectations. During the research, it became clear that it is also very important to constantly pay attention to demand planning and not to forget to develop and establish loyal relations with the customer; it is necessary to focus on the possible optimal use as a sales team management tool. Based on the opinion of the experts participating in the study, a basic list of the importance of investments in the creation and development of the technological infrastructure of the transport company should be drawn up.

Due to the limits of the selected study, important criteria such as $CO_2$ emissions and the importance of artificial intelligence were not analysed in full detail. $CO_2$ emission is understood as a general indicator of pollution, which by default includes other hazardous substances released by an internal combustion engine powered by diesel fuel, and this indicator is also important when trying to evaluate the efficiency of technologies—how efficiently and properly energy is used and created with its help. In addition, the decreasing amount of $CO_2$ may indicate that companies are using alternative fuels or an alternative type of energy. At the same time, the problem of $CO_2$ emission is actualised in society, and a stable attitude is formed in the public space that minimisation of $CO_2$ emission is a necessity connected to nature protection, new lifestyles, and values. Being green is welcome, favourable, and commendable. It is a positive attitude that creates a favourable attitude towards the individuals and businesses that also practice this attitude. Therefore, it can be said that the $CO_2$ emission element is also related to the marketing complex. It is unlikely that consumers will favourably receive any new technology if it does not emphasise an element of greenness expressed through the minimisation or at least positive management of $CO_2$ emissions. Positive $CO_2$ emission management can be understood as the recalculation of the amount of $CO_2$ emissions per unit of cargo during the transportation of goods, proving that the emission per unit of cargo has been reduced to a minimum.

The issue of artificial intelligence is discussed in the literature but requires additional analysis. One of the more essential analysis criteria could be "human–machine" interaction, which comprises the issues of trust, competencies, and decision making. It can be assumed that too much emphasis is placed on artificial intelligence in technological processes; it can be understood as a challenge for personnel to make final decisions and evaluate possible solutions. The issue of artificial intelligence is likely to trigger additional debates regarding the competencies of personnel working with it and their role in the technological process. Blind trust in the decisions made (proposed) by artificial intelligence can create conditions for the degradation of human competencies and abilities. Many questions may arise when the issue of responsibility for decisions made by artificial intelligence arises. Although artificial intelligence technological solutions are being developed, they are popular because they reduce the cost of customer service and speed up processes, but ethical, moral, and liability issues remain. On the other hand, it can be assumed that widely used but low-tech artificial intelligence tools for customer service are quickly recognised by customers and negatively affect them.

Therefore, it is necessary to conclude that these two criteria—$CO_2$ emission and artificial intelligence—could be analysed separately, focusing on their importance in the interaction of technology and marketing.

This article's authors' study results are an assessment of how digital technologies affect the transport industry. The study findings can serve as a basis for formulating transport policy, which is part of the digital transformation strategy of the transport industry.

## 6. Conclusions

1.  After analysing the sources of scientific literature, it can be assumed that the attractiveness and availability of new technologies also encourage the development and broader application of these technologies in practice. The analysed information and communication technologies are associated with an essential factor of mobility, which, by its nature, is inseparable from the activities of transport and logistics companies. Analysing the sources, a possible dependence was observed—technological development is possible and encouraged through the devices used by the user. It can be said that there is a dependency—when marketing promotes technologies and their wider use, the development of technologies changes marketing, which allows market research in a completely different way to determine the mechanisms of impact on the user and the means of market formation. The possible trend is that the marketing of the future will not be so directly tied to a person or a classic survey but will be connected with information from big data or cloud data arrays.

2.  A multi-criteria research method was chosen to evaluate the influence of marketing on the technological development of transport companies and the intensity of technological development and the quality assurance of transport companies' services. The chosen method is flexible, convenient for decision makers, and can assess the consistency of the opinion of several experts. The number of interviewed experts and their competence was sufficient to ensure the reliability of the research and its results. It can be said that the obtained research results meet the requirements of the methodology and are reliable, and the conclusions based on them are reasonable.

3.  Based on experts' assessments and calculations, the main list of the importance of transport companies' marketing strategy in the creation and development of technological infrastructure should include the following components: first, the conditions must be created, both technological and organisational, to perform automated tracking of customer choices. Second, marketing and technological infrastructure elements must focus on a more personalised, customised process that meets customer expectations. Third, there is a need to pay constant attention to demand planning. Fourth, there is a need to develop and establish a loyal relationship with the customer. Fifth, it is necessary to focus on the possible optimal use as a sales team management tool. According to experts' assessments and calculations, the main list of the importance of investments in the creation and development of the technological infrastructure of a transport company should be made firstly by focusing on quick troubleshooting, secondly by reducing the amount of carbon dioxide, and thirdly by ensuring the continuity of the cargo transportation process. Other elements of importance are associated with the need to improve road traffic safety and orientation towards the introduction of digital transformation.

4.  The discussion section discusses the limitations of the study and unexplored or insufficiently analysed areas that may be significant in the context of the researched topic. This is the importance and role of artificial intelligence in technological processes, especially in customer service and the concept of $CO_2$ emission and the management of this emission to achieve energy efficiency and environmental aspects related to marketing activities.

**Author Contributions:** Conceptualisation, M.I. and K.V.; methodology, K.V.; software, D.B.; validation, D.B., M.I., and K.V.; formal analysis, M.I.; investigation, K.V.; resources, K.V.; data curation, K.V.; writing—original draft preparation, M.I.; writing—review and editing, D.B.; visualisation, K.V. and D.B.; supervision, K.V.; project administration, D.B.; funding acquisition, K.V. All authors have read and agreed to the published version of the manuscript.

**Funding:** This research received no external funding.

**Institutional Review Board Statement:** The study was conducted in accordance with the Declaration of Helsinki, and approved by the Institutional Review Board (Ethics Committee) of Vilnius Gedimino Technical University (protocol code 10.6-07-10.21-8634; 23 August 2022).

**Informed Consent Statement:** Informed consent was obtained from all subjects involved in the study.

**Conflicts of Interest:** The authors declare no conflict of interest.

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
