# Peer review of "A Study of the Relationship between Marketing and Investment in Technology Development in Transport Company"

_sustainability, doi:10.3390/su141912927_

Round 1
Reviewer 1 Report
It's a relevant topic for many logistics service providers. It is one of the first (good) articles to investigate the impact of marketing on the technological development of the transport company and the intensity of the technological development, and the quality assurance of the transport company’s services.
Unclear is who the experts interviewed were. Industry experts? From which segments of the transport industry?
However, the step on page 8 and figure 3 toward criteria lacks a theoretical basis. How did the researchers get to this list? Is this list complete? What is the (value chain or business model) framework behind these criteria? This needs more exploration. It now only states: "The study involved 8 experts from different transport enterprises. The experts had 5 to 8 years of experience in managing logistics and transport enterprises".
This kind of publication could also be stronger if linked to grey literature on the transport industry e.g. the yearly CSCMP reports made by universities: 33rd Annual State of Logistics Report.
Author Response
Thanks, you very much. The text of manuscript has been improved.

Reviewer 2 Report
The topic of the manuscript is relevant, and its content may be of interest to potential readers. However, there are lot of gaps and shortcomings that authors need to remove before a manuscript can be published:
1. Regarding the abstract:
1.1. It is advisable to increase the volume of the annotation. It is necessary to pay more attention to the obtained results. In particular, it is worth mentioning the digital transformation.
1.2. I suggest adding the phrase "digital transformation" to the list of keywords.
2. Regarding section 1:
2.1. This section is too small in size. It is necessary to more fully describe the relevance of the research topic.
2.2. It is necessary to submit a more detailed presentation of the obtained results and their scientific novelty. This can be done after formulating the research objective. For example, the following can be noted: "In the process of achieving the study goal, a number of results were obtained, which contain elements of scientific novelty. First, there was …… Second, ……Third, …” Based on the content of the manuscript, it contains elements of scientific novelty that have both theoretical and applied nature.
2.3. The last introduction paragraph should describe the manuscript structure. It is advisable to start this paragraph as follows: “Further material is divided into several parts. Thus, in Section 2……. Section 3 presents……”.
3. Regarding the structure of the manuscript. I propose the following structure:
1. Introduction
2. Literature review
2.1. Technological development in the transport company
2.2. Marketing impact on technological development
3. Methodology
4. Empirical analysis (starting from line 292)
5. Discussion
6. Conclusions
4. Regarding the literature review:
4.1. The literature review is very monotonous. A critical analysis and comparison of the opinions of different scientists is needed.
4.2. Is it appropriate to submit Table 1? After all, this information concerns only one country (the Republic of Uzbekistan).
4.3. The content of the sections does not exactly correspond to their names. For example, the following is noted in lines 150-152: “Başyazıcıoğlu and Karamustafa [12] mentioned that technological developments allow the development of marketing activities, simplify and speed up with new tools and methods”. However, the title of section 3 implies the opposite effect ("Marketing impact on technological development").
5. Regarding the methodology:
5.1. In lines 248-249, the following is noted: "A multi-criteria approach was chosen to assess the impact of the technological development of the transport enterprises on the marketing strategy". This does not correspond to the purpose of the research (lines 34-36): "The purpose of the research is to investigate the impact of marketing on the technological development of the transport company and the intensity of the technological development and the quality assurance of the transport company's services".
5.2. The mechanism of marketing influence on the technological development of transport companies remains unclear. Part of this mechanism is described in Figure 3. However, this is not enough. What are the influencing factors? What is the resulting indicator (indicators)?
5.3. It is appropriate to clearly list the main areas of technological development of transport companies. How to assess the level of this development?
5.4. Probably, in the manuscript it would be necessary to assess the degree of influence of marketing activities of transport companies on the level of their technological development. However, it is doubtful that this can be done using multi-criteria ranking.
6. Regarding the empirical analysis:
6.1. The obtained results do not fully correspond to the purpose of the study. I would like to see clear results of the assessment of the impact of marketing on the technological development of transport companies.
6.2. Is the sample of transport companies representative?
6.3. It seems that the titles of tables 3 and 4 do not quite correspond to the topic of the study.
6.4. The following is noted in lines 266-267: "The research process is based on the statement (hypothesis) ‒ the technological development strategy of the transport company depends on the market trends." It is necessary to write about whether this hypothesis turned out to be correct.
7. Regarding the section “Model marketing impact on technological development of transport company”:
7.1. I suggest removing this section.
7.2. Figure 3 can be submitted in the "Methodology" section.
7.3. The material given in lines 427-487 is rather trivial.
8. Concerning the discussion of results and conclusions.
8.1. The discussion needs to be improved significantly. In particular, it is necessary to compare the results obtained by the authors with known results, that is, obtained by other scientists.
8.2. In the conclusions, the authors pay more attention to the technological infrastructure. However, technological infrastructure is not exactly the same as technological development.
8.3. It is desirable to indicate directions for further research.
9. Regarding design, grammar and style:
9.1. All symbols in the formulas must be decoded. Please check carefully.
9.2. The authorship of tables and figures can be better indicated in the notes.
9.3. You need to check your grammar. Some sentences are not very well worded. For example, in lines 71-74.
9.4. In some places of the manuscript, the style should be improved, making it more scientific. In particular, this applies to the material presented in lines 427-487 and the description of research methods. This text sometimes looks like the text of a lecture rather than part of a scientific article.
I think it is appropriate to acquaint the authors with these comments, suggestions and questions. I hope that such acquaintance help to improve the quality of the manuscript, which is expected to be published in such a high-ranking journal as "Sustainability".
Author Response

(The authors gave the same response as above.)

Reviewer 3 Report
The subject matter is very important. The paper is interesting and overall nicely done. However, the conclusions that there is a need for a personalised experience and a need to build brand loyalty, as well as reduce CO2 seem all to be rather obvious in the sense of "current conventional wisdom." Now, conventional wisdom isn't always right, and studies are needed to verify it. I would very much like to see this addressed an expanded in the conclusions section so that it comes across as a novel idea and does not result in a reader possibly saying, "So what? I already know that." With what I have seen in the paper, I think this should be easily doable.
Also, can you address negative issues of virtual chat bots and such? A lack of human contact is a problem for many. However, only consider this if it will not result in taking you away from the central theme.
Author Response

(The authors gave the same response as above.)

Round 2
Reviewer 2 Report
The authors have taken into account a number of my remarks. Though, there are still some shortcomings in the manuscript, some of which arose after the changes made by the authors:
1. Main remarks:
1.1. Considering section 4, two main results obtained by the authors can be distinguished:
1) the list of importance of the main criteria for the impact of technological development of the transport company's intensity of investments in the creation and development of technological infrastructure;
2) the list of importance of the main criteria for the impact of the marketing strategy of a transport company on the implementation of technological development.
I have the following questions:
What does the first result have to do with the topic of the study?
What is the mechanism of the impact of the marketing strategy of a transport company on the implementation of technological development? Perhaps the needs of marketing determine the improvement of technological processes?
In section 3, authors should provide answers to these questions. Or you need to adjust the title of the manuscript, the purpose of the study, etc.
1.2. In the "Discussion" section, you should discuss your own results, and not the results obtained by other scientists. In this section you need to answer the following questions:
What explains the obtained results?
What can be considered the advantages of this research in comparison with analogues?
What are the reasons for these benefits?
What are the limitations of this study? (You can take some material from section 6).
What are the shortcomings of this study?
What are the prospects for further research on this topic? (You can take some material from section 6).
2. Other remarks:
2.1. The text in lines 73-82 should be deleted. It does not refer to the scientific novelty of the research results.
2.2. On lines 86-88, provide information about sections 5 and 6.
2.3. Subsection 2.2 should be called something like this: "The relationship between marketing and technological development".
2.4. It is necessary to provide more complete names of figures 1 and 2.
2.5. The last sentence in section 6 (line 657) should be removed. It is better to talk in more detail about the validity of the hypotheses in section 5. At the same time, these hypotheses must be formulated both in the previous text and in section 5. Highlight these formulations in the text.
Author Response
Response: Thanks, you very much. The text of manuscript has been improved

Round 3
Reviewer 2 Report
The text of the manuscript still contains a number of shortcomings, namely:
1. The following title of the article is given in lines 2-3: "A study of the relationship between marketing and investment in technology development in a transport company". In the company or in companies?
2. It is not very good to do a literature review in the introduction (lines 41-62). But if you do, then you need to consider scientific works purely on the topic of the article (about marketing and technological development of transport companies). It is necessary to replace at least sources 3 and 4 (lines 59-62).
3. In lines 63-65 it is noted: "The purpose of the research is to investigate the impact of marketing on the technological development of the transport company and the intensity of the technological development, and the quality assurance of the transport company's services". The specified purpose does not exactly correspond to the corrected title of the article. I propose to formulate something like this: "Taking into account the above, the purpose of this paper is the study of the relationship between marketing and investment in technology development in transport companies." Or the authors can take as a basis the previous goal, but then the word "investment" must appear.4. In lines 65-66 it is noted: "The relevance of the research..." I believe that it should be written approximately as follows: "The peculiarity of this research..."
5. In lines 202-203 it is noted: "Based on the opinion of the scientists as mentioned above, the development of technology..." Is this true? It is better to write like this: "It can be assumed that technological development..."
6. Line 309 states: "Technological developments affect all elements of the marketing mix". It is better to write like this: "Thus, technological developments affect all elements of the marketing mix".
7. In lines 388-390 it is noted: "During the research, a hypothesis was put forward that the marketing strategy influences the technological development of Lithuanian transport companies and their intensity of investment". But in section 5 "Discussion" we are talking about not one, but two hypotheses (lines 591-594).
8. It also seems to me that it is inappropriate to place the formulation of the hypothesis in lines 362-363. It is not necessary to highlight in bold. You can italicize, but only the formulation of hypotheses.
9. In line 390, "Eight logistics specialists from Lithuanian logistics and..." It is better to write not "eight", but "several", so as not to repeat line 405.
10. In lines 523-524 it is noted: "Figure 3 presents the Model marketing impact on the technological development of transport companies". It is better to write like this: "Taking into account the above, it is possible to propose a model marketing impact on the technological development of transport companies. This model is shown in Figure 3”.
11. Line 544 states: “Model marketing impact on the technological development…” It is better to write as follows: “As can be seen from Figure 3, the model marketing impact on the technological development…”
12. Lines 555-590 should be deleted. Or transfer them to the literature review.
13. In lines 610-611, it is stated: “The advantage of the research Marketing Impact on Technological Development Transport Company is that…” It is better to write as follows: “The advantage of the performed research is that…”
14. Conclusions need to be significantly improved. They may be smaller in scope, but specific. It is possible to make three paragraphs: regarding the conducted literature review; regarding the proposed methodology; regarding the results of the empirical analysis.
15. All symbols in the formulas must be decoded immediately. In particular, look at the symbols in formulas (1) to (4).
16. Why are some words underlined (in particular, in lines 442, 443, 547)?
17. I urge the authors to read the text carefully in order to avoid typos and grammatical errors. In particular, look at lines 424, 691, etc.
Author Response
Dear Reviewer,
We appreciate your precious time in reviewing our paper and providing valuable comments. The authors have carefully considered the comments and tried our best to address every one of them. We hope the manuscript after careful revisions meet your high standards. All modifications in the manuscript have been highlighted.
